# Red Blood Cell Alloimmunization and Its Associated Factors among Chronic Liver Disease Patients in a Teaching Hospital in Northeastern Malaysia

**DOI:** 10.3390/diagnostics13050886

**Published:** 2023-02-25

**Authors:** Siti Zaleha S. Abdullah, Mohd Nazri Hassan, Marini Ramli, Marne Abdullah, Noor Haslina Mohd Noor

**Affiliations:** 1Department of Haematology, School of Medical Sciences, Universiti Sains Malaysia, Kubang Kerian 16150, Malaysia; 2Transfusion Medicine Unit, Hospital Universiti Sains Malaysia, Kubang Kerian 16150, Malaysia

**Keywords:** chronic liver disease, alloantibody, alloimmunization, blood transfusion

## Abstract

Red blood cell (RBC) alloimmunization is an important complication of blood transfusion. Variations in the frequency of alloimmunization have been noted among different patient populations. We aimed to determine the prevalence of RBC alloimmunization and associated factors among chronic liver disease (CLD) patients in our center. This is a case-control study involving 441 patients with CLD who were being treated at Hospital Universiti Sains Malaysia and subjected to pre-transfusion testing from April 2012 until April 2022. Clinical and laboratory data were retrieved and statistically analyzed. A total of 441 CLD patients were included in our study, with the majority being elderly, with the mean age of patients 57.9 (SD ± 12.1) years old, male (65.1%) and Malays (92.1%). The most common causes of CLD in our center are viral hepatitis (62.1%) and metabolic liver disease (25.4%). Twenty-four patients were reported to have RBC alloimmunization, resulting in an overall prevalence of 5.4%. Higher rates of alloimmunization were seen in females (7.1%) and patients with autoimmune hepatitis (11.1%). Most patients developed a single alloantibody (83.3%). The most common alloantibody identified belonged to the Rh blood group, anti-E (35.7%) and anti-c (14.3%), followed by the MNS blood group, anti-Mia (17.9%). There was no significant factor association of RBC alloimmunization among CLD patients identified. Our center has a low prevalence of RBC alloimmunization among CLD patients. However, the majority of them developed clinically significant RBC alloantibodies, mostly from the Rh blood group. Therefore, phenotype matching for Rh blood groups should be provided for CLD patients requiring blood transfusions in our center to prevent RBC alloimmunization.

## 1. Introduction

Globally, 1.5 billion people have been diagnosed with chronic liver disease (CLD), with an estimated incidence of cirrhosis in Southeast Asia being 23.6 per 100,000 [1]. CLD of any cause is frequently associated with hematological abnormalities, including anemia which is due to multiple causes: reduced red blood cell (RBC) lifespan, hemolytic anemia, and bleeding due to varices, thrombocytopenia, or coagulation factor deficiency [2]. Current guidelines recommend blood transfusions to a goal of 7–8 g/dL in patients with cirrhosis [3]. As the survival of CLD patients has greatly increased with the advancement in medical treatment, they may require repeated blood transfusions throughout the course of their illness, which puts them at high risk for RBC alloimmunization [4].

RBC alloimmunization is a common, undesirable outcome of blood transfusion that occurs as a response of the recipient’s immune system to foreign RBC antigens, particularly in chronically transfused patients [5]. These antibodies may be clinically significant, leading to hemolytic transfusion reaction (HTR) or hemolytic disease of the fetus and newborn (HDFN) [6]. RBC alloantibodies may also cause some technical issues and result in difficulty and delay in providing compatible blood [7]. Therefore, RBC alloantibody detection and identification are critical in transfusion practice to provide antigen-negative blood to patients.

Bajpai et al. reported that patients with underlying CLD had an overall statistically significant impact on the rate of alloimmunization [4]. Few studies on multiple RBC transfusion and alloimmunization showed a significant association between the mean number of packed red blood cells (PRBCs) transfusions and the number of alloantibodies formed because a higher number of transfusions will increase the possibility of encountering a foreign antigen [8,9]. A better understanding of the risk factor of RBC alloimmunization is needed to optimize prevention strategies and increase transfusion safety [6]. Many previous studies have concluded that preventing RBC alloantibody formation may prevent HTR, increase life expectancy, and reduce the amount of blood transfusion required in patients who need frequent blood transfusions [6,10].

In Malaysia, multiple ethnicities have caused genetic heterogeneity among the population, which in turn leads to a wide variation of antibody specificity [6]. Previous studies in Malaysia showed the incidence of irregular antibodies detected in antibody-screening positive cases ranged from 0.76% to 1.4% of pre-transfusion samples [11]. There were a limited number of studies reported on RBC alloimmunization in CLD patients, with the incidence ranging from 5.3% to 7.4% [4,12]. However, there is no previous study on RBC alloimmunization in CLD patients among the Malaysian population. Thus, this study aims to identify the frequency and specificity of RBC alloimmunization among CLD patients in our center. The findings of this study will contribute to a better understanding of the risk factors associated with the development of RBC immunization in CLD patients. RBC alloimmunization rates vary with disease status, the benefit of extended or selected antigen matching as preventive measures of RBC alloantibody formation in CLD patients for transfusion management will be determined.

## 2. Materials and Methods

This is a case-control study involving 441 adult patients (18 years old and above) with CLD who were being treated at Hospital Universiti Sains Malaysia (USM) and subjected to pre-transfusion testing from April 2012 until April 2022. Patients under 18 years old or those with comorbidities such as thalassemia, sickle cell disease, and hematological malignancy were excluded from this study. Twenty-six patients were noted to have positive antibody screening, in which two of them were noted to be due to autoantibody. The study was approved by the Human Research Ethics Committee USM (HREC) using protocol code USM/JEPeM/21030228.

Clinical and laboratory data were retrieved from the patient’s medical record and blood bank information system. Patient’s sociodemographic including age, race, gender, smoking habit, and alcohol consumption, clinical data including causes of CLD, and laboratory data including ABO/Rhesus D (RhD) blood group, transfusion history, type of blood product transfused, antibody screening and identification, antibody specificity, RBC phenotyping, liver function test (LFT), hemoglobin (Hb) level and platelet count were included for analysis. The RBC alloantibody may be either clinically significant or insignificant. A clinically significant alloantibody is defined as an antibody that can cause shortened survival of RBCs, leading to HTR or HDFN [13].

About 3 mL of blood samples were collected in ethylene diamine tetra acetic acid (EDTA) tube from each patient and tested for ABO and RhD grouping and RBC antibody screening. Antibody screening was performed using a commercial three-screening cell panel (ID-Diacell I-II-III, Asia, Bio-Rad, Cressier FR, Switzerland) at 37 °C using low ionic strength solution (LISS) and anti-human globulin (AHG). Sample with positive antibody screening proceeded with antibody identification using an eleven-cell panel, using LISS and enzyme-treated cells (papain) at 37 °C and AHG phase. The entire test was performed using a commercialized cell panel by microcolumn gel agglutination method (ID-DiaPanel, Bio-Rad). RBC phenotyping toward the antibody specificity except Mia was performed using commercial reagents based on the manufacturer’s recommendation to confirm the presence of the alloantibody identified. The interpretation of anti-Mia is just based on the positive reaction at Cell III antibody screening. The Mia phenotype was not confirmed due to the unavailability of Mia antisera.

Data were analyzed using Statistical Package for Social Sciences (SPSS) Statistics (IBM Corp. Released 2013. IBM SPSS Statistics for Windows, Version 26.0., IBM Corp., Armonk, NY, USA). Data obtained were expressed as mean and standard deviation (SD) for numerical, and the categorical variables were described in frequency and percentage (%). The prevalence of RBC alloimmunization and the alloantibody specificities were described using descriptive statistics. For the determination of risk factor for RBC alloimmunization, simple (SLR) and multiple logistic regression (MLR) is used. *p* < 0.05 was considered to indicate statistical significance.

## 3. Results

A total of 441 patients were included in our study, with the majority being elderly. The mean age of patients was 57.9 (SD12.1) years old. Most of the patients were male (65.1%), Malays (92.1%), and had a history of transfusion (54.9%). The most common causes of CLD in our center were due to viral hepatitis (62.1%), followed by metabolic liver disease (25.4%), cryptogenic liver disease (5.9%), and other causes, as shown in Figure 1. The details of descriptive data are illustrated in Table 1.

### 3.1. Prevalence and Specificity of RBC Alloantibody among CLD Patients

We found that 24 patients were detected to have alloantibody with an overall prevalence of 5.4%. The mean age of alloimmunized patients was 56.2 (SD ± 12.4), with the majority being male (54.2%), Malay (95.8%), A blood group (37.5%), CLD secondary to viral hepatitis (58.3%) and had a history of transfusion (58.3%). All the alloimmunized CLD patients were RhD-positive (Table 1). Among the 24 patients with alloantibodies, the majority had a single antibody (83.3%), and only 16.7% of patients had multiple antibodies detected. More than half of the alloimmunized patients (54.2%) had antibodies against the Rh system. The rate of clinically significant alloantibodies was 58.3% (14/24 patients). The frequency and specificities of the various alloantibodies identified are shown in Table 2.

A total of 28 RBC alloantibodies had been detected from those 24 alloimmunized CLD patients. Among these, the most common alloantibody identified belonged to the Rh blood group (57.1), which mainly contributed by anti-E (35.7%) and anti-c (14.2%), followed by anti-Mia (17.8%) from the MNS blood group (Figure 2).

The details of clinical data for all alloimmunized CLD patients are summarized in Table 3. We observed that most of the patients with a previous history of transfusion developed a clinically significant alloantibody (*n* = 12), and the majority of them received <10 units of packed cell transfusion (*n* = 12).

### 3.2. Factors Associated with RBC Alloimmunization among CLD Patients

The factors associated with RBC alloimmunization among CLD patients that had been analyzed in this study include age, gender, ethnicity, smoking habit, Hb level, platelet count, ABO/RhD blood group, history of transfusion, and the number of transfusions. This study showed no significant association of RBC alloimmunization with all selected factors. Those factors with *p* < 0.25 (platelet count, WBC count, and the number of platelet transfusions) in the univariate analysis were included in the multivariate analysis.

Although the results were not statistically significant, we observed that the alloimmunization rate was higher in females, with 7.14% (11/154) compared to 4.53% (13/287) in males. In this study, the mean number of PRBC units transfused in alloimmunized patients was 2.75 units, which was not significantly different from the non-alloimmunized patients (2.31 units). The statistical analysis is shown in Table 4.

## 4. Discussion

Our study reported the prevalence of RBC alloimmunization among CLD patients was 5.4% which was within the reported range of 5.2% to 7.4% by other studies on RBC alloimmunization in patients with liver disease among Asian populations [4,12]. Previous studies among the transfused patient population in Malaysia by Nadarajan et al. and Yousuf et al. showed a low prevalence of RBC alloimmunization rate of 1.6% and 0.8%, respectively [11]. A higher alloimmunization rate was seen in transfusion-dependent thalassemia patients, which was approximately 10%, and about 20% among sickle cell disease patients [14,15,16]. Differences in the alloimmunization rate among the studied population could be attributed to numerous factors such as patient demography, comorbidities, number of transfusions, local transfusion practice, the sensitivity of the test method (antibody screening and identification), and heterogeneity of the population studied [4,11,17,18,19]. The reason why only a subset of patients, known as “responders”, form alloantibodies to donor RBCs despite multiple exposures via transfusion or pregnancy is complex and poorly understood [20]. Karafin et al. reported age, gender, race, and certain hematological and autoimmune diseases (including sickle cell disease or trait, systemic lupus erythematosus, rheumatoid arthritis, and myelodysplastic syndrome) were associated with being a responder [20].

### 4.1. RBC Alloantibody Specificity among Alloimmunized CLD Patients

Our data showed that the majority of alloimmunized patients had single rather than multiple alloantibodies. Among our patients, the most common alloantibodies found belonged to the Rh system (54.2%). Anti-E is the most common alloantibody identified, accounting for 35.7% of the patients, followed by anti-Mia (17.9%) and anti-c (14.3%). This was similar to the previous reports of RBC alloimmunization among transfused patients in Malaysia as well as other regions such as Korea, India, Pakistan, the US, and Brazil, which found anti-E as the most common alloantibody identified [4,12,15,21,22,23,24,25]. These findings indicate that the E antigen is highly immunogenic, and patients with negative E antigen were prone to be sensitized following blood transfusion [17,26]. Furthermore, individual with negative E antigen is common among the Malay population, accounting for about 77% of the population [27]. In addition, anti-E can also be a naturally occurring antibody that may present without a sensitizing event [13,28,29]. However, we could not distinguish between naturally occurring or immune anti-E as all the reactions observed with the cells panel were more intense in the enzymatic medium than in the presence of AHG. In addition, we did not proceed with dithiothreitol (DTT) treated serum due to the unavailability of the reagent in our center. We found that only one out of ten alloimmunized patients with anti-E (Table 4) did not have any history of transfusion and/or pregnancy, which could be considered naturally occurring anti-E. Kim et al. reported that among Korean patients with CLD, the most common alloantibody found was anti-E (45%), followed by anti-c (17%), which was similar to our finding [12]. All Rhesus group antibodies are clinically significant, with variable presentation ranging from mild to severe hemolysis [13]. Our current transfusion practice does not provide the Rh phenotype-matched blood to all patients but only for transfusion-dependent thalassemia and RhD negative, which can explain the Rh alloimmunization in most of our CLD patients. This also explains the need for extended Rh phenotype-matched RBCs in patients requiring repeated transfusions to prevent Rhesus alloimmunization [11].

Our study found anti-Mia as the second most common alloantibody detected. The anti-Mia antibody was described as the most common alloantibody in several studies among transfusion recipients of Asian populations [11,30,31,32]. Few studies reported that anti-Mia antibody is common in Asian populations, including Malaysia, but very rare among Caucasians [4,5,30,31]. Musa et al. reported that 15 of 156 donors (9.6%) among Malaysians were positive for Mia antigen compared to an incidence of less than 0.01% among Caucasians [27,31]. Most of these antibodies are IgM and tend to react best at cold temperatures. However, there are some that are IgG, reactive at 37 °C, and clinically significant [31]. There have been few case reports of clinically significant anti-Mia antibodies resulting in HTR and HDFN [33,34]. All these studies suggest the need for proper detection of anti-Mia antibodies during pre-transfusion testing, especially among Asian populations [33,34].

In this study, only four patients were found to have multiple antibodies, with three of them developing anti-E and anti-c. These patients had a history of multiple PRBC and platelet transfusions. This result was comparable to the results from other studies in which anti-E and anti-c were found to be the most common combination of multiple alloantibodies [5,11,22]. It was reported that E and c were among the 10 most potent antigens to stimulate RBC alloimmunization, and these can stimulate multiple antibodies in antigen-negative patients [35]. In addition, few studies described the probability of additional antibody formation increases approximately three-fold in alloimmunized patients [22,36]. This indicates that patients with a single alloantibody requiring regular blood transfusions have a higher risk of developing multiple alloantibodies compared to patients without alloantibody. Based on this observation, many authors suggest the use of extended phenotypic matching of PRBCs for patients with regular blood transfusions [14,15,17,23,37].

Kell is one of the high antigenicity antigens, and anti-Kell is a clinically significant alloantibody that is common among Caucasians. However, we did not observe anti-K among our CLD patients, likely due to the low prevalence of K antigen (0.14%) in our population compared to Caucasians (9%) [12,31].

### 4.2. Factors Associated with RBC Alloimmunization in CLD Patients

This study showed differences in the rates of RBC alloimmunization among various liver diseases. However, they were not statistically significant. The highest rates of RBC alloimmunization were detected among patients with autoimmune hepatitis (11.1%), followed by metabolic liver disease (7.1%) and viral hepatitis (5.1%). Our findings were similar to previous studies that showed a higher rate of RBC alloimmunization among patients with the autoimmune disease compared to other causes of CLD [4,6,20,38]. Patients with autoimmune diseases such as systemic lupus erythematosus are more likely to become RBC alloimmunized, even though they are rarely transfused [7]. This could be explained by an enhanced immune response in a patient with an autoimmune disease that may favor RBC alloimmunization based on an immunogenetic study [4,20,38]. Autoimmune disease is also known as a marker of a potential antibody ‘responder’ pertaining to PRBC transfusion [20]. Alloimmunization was not seen in patients with alcoholic liver disease in our study. This result is in keeping with a previous study in India that showed a low rate of alloimmunization among patients with alcoholic liver disease (4). This may be due to the suppression of the immune response as a result of depleted T cells, B cells, natural killer cells, and monocytes in patients with chronic alcoholism [4,39].

Even though our result showed no significant association between gender and risk of RBC alloimmunization, we observed that females had a greater frequency of RBC alloimmunization compared to males, with a male-to-female ratio of 1:1.6. This finding was similar to other studies that showed a higher rate of alloimmunization among females compared to males due to greater exposure to immunizing events such as pregnancy and transfusions [4,6,17]. Yusoff et al. concluded that the history of pregnancy had a significant association with the development of RBC alloantibody [17]. Pregnancy is a known trigger factor of RBC alloimmunization in females due to exposure to fetal RBCs during pregnancy and around the time of delivery. Each pregnancy carries the risk of RBC alloantibody formation. This is because when exposed to paternal foreign red blood cell antigens during pregnancy, the immune system produces antibodies against the respective antigen. [17]. However, the majority of women do not become alloimmunized after this exposure because the immunogenicity of the RBC antigen appears to be one of the critical factors in the development of RBC alloimmunization in pregnancy [6].

The mean number of PRBC units transfused among alloimmunized CLD patients was 2.8 units, which was not much different from the non-alloimmunized CLD patients (2.3 units). However, result from previous studies among transfused patients showed the number of PRBC units transfused was significantly higher in alloimmunized patients compared to the non-alloimmunizaed patients [4,6,11,18]. These discordant results were probably due to the low number of blood transfusions among CLD patients in this study, which may have contributed to the low prevalence of RBC alloimmunization. A previous study on the analysis of the transfusion characteristics revealed that the rate of RBC alloimmunization was significantly associated with the number of PRBC received, the number of transfusion episodes, and the number of donor exposures [4,18,22].

Our study involved only a small number of alloimmunized patients due to the limited number of cases available within the study period. There is also a possibility that we missed a low titer antibody or some other antibody that had not been detected by our screening panel. Therefore, the findings need to be inferred with caution since they might not be representative of the reference population. A large-scale prospective study should be conducted in which antibody detection is carried out at specific time intervals after blood transfusion.

## 5. Conclusions

This study showed a low prevalence of RBC alloimmunization among CLD patients treated in our center. However, the majority of them developed clinically significant RBC alloantibodies, mostly from the Rh blood group. Therefore, CLD patients requiring blood transfusion in our center must at least be phenotyped for the Rh system and supplied with Rh phenotype-specific blood for transfusion to prevent Rhesus alloimmunization. For patients with anti-Mia, Mia antigen matching also could be considered for laboratory with good financial and staffing since it involves phenotyping not only for the recipient but also the donors.

## Figures and Tables

**Figure 1 diagnostics-13-00886-f001:**
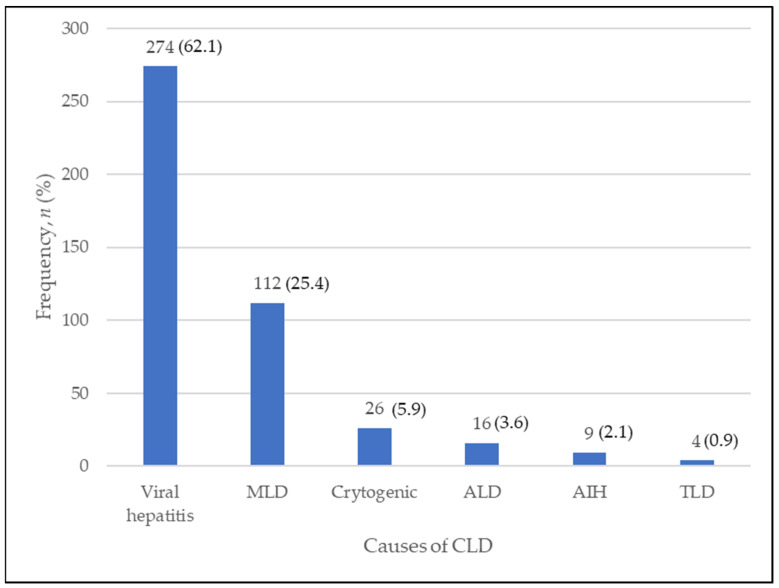
The distribution of patients according to the causes of CLD. CLD = chronic liver disease; MLD = metabolic liver disease; ALD = alcoholic liver disease; AIH = autoimmune hepatitis; TLD = toxic liver disease.

**Figure 2 diagnostics-13-00886-f002:**
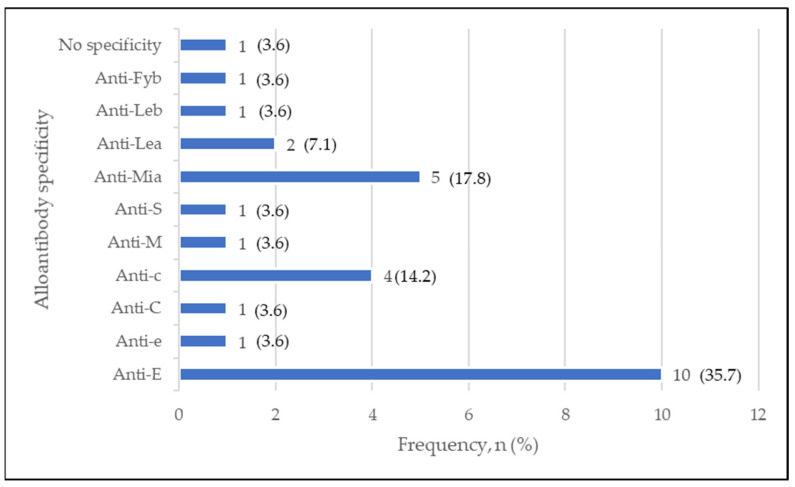
Frequency of 28 RBC alloantibodies detected in CLD.

**Table 1 diagnostics-13-00886-t001:** Descriptive characteristics of patients with CLD (*n* = 441).

Variables	All Patients, *n* (%)	Alloimmunized Patients, *n* (%)	
		Yes (*n* = 24)	No (*n* = 417)
Age (years) *	57.9 (±12.1)	56.2 (12.4)	58.01 (12.1)
Gender			
Male	287 (65.1)	13 (54.2)	274 (65.7)
Female	154 (34.9)	11 (45.8)	143 (34.3)
Race			
Malay	406 (92.1)	23 (95.8)	383 (91.8)
Non-Malay	35 (7.9)	1 (4.2)	34 (8.2)
Causes of CLD			
Viral hepatitis	274 (62.1)	14 (58.3)	260 (62.4)
MLD	112 (25.4)	8 (33.3)	104 (24.9)
Cryptogenic	26 (5.9)	1 (4.2)	25 (6.0)
ALD	16 (3.6)	0 (0)	16 (3.8)
AIH	9 (2.0)	1 (4.2)	8 (1.9)
TLD	4 (0.9)	0 (0)	4 (1.0)
ABO group			
O	139 (31.5)	6 (25.0)	133 (31.9)
A	124 (28.1)	9 (37.5)	115 (27.6)
B	138 (31.3)	8 (33.3)	130 (31.2)
AB	40 (9.1)	1 (4.2)	39 (9.3)
RhD group			
+ve	438 (99.3)	25 (100)	414 (99.3)
−ve	3 (0.7)	0 (0)	3 (0.7)
WBC (10^9^/L) *	9.3 (13.8)	6.9 (3.4)	9.5 (14.1)
Hb (g/dL) *	8.9 (2.3)	8.5 (2.1)	8.9 (2.3)
Plt (10^9^/L) *	130.2 (93.2)	102.2 (89.4)	131.8 (93.3)
History of tx			
Yes	242 (54.9)	14 (58.3)	228 (54.7)
No	199 (45.1)	10 (41.7)	189 (45.3)
PRBC tx (U) *†	2.3 (3.7)	2.8 (4.5)	2.3 (3.6)
Plt tx (U) *†	1.5 (4.03)	2.9 (5.9)	1.4 (3.9)
Smoking (Yes/No)			
Yes	176 (39.9)	7 (29.2)	169 (40.5)
No	265 (60.1)	17 (70.8)	248 (59.5)
Alcohol (Yes/No)			
Yes	35 (7.9)	0 (0)	35 (8.4)
No	406 (92.1)	24 (100)	382 (91.6)

* = mean (SD±); † = among transfused patients (*n* = 242); M = male; F = female; Non-M = Non-Malay; CLD = chronic liver disease; MLD = metabolic liver disease; ALD = alcoholic liver disease; AIH = autoimmune hepatitis; TLD = toxic liver disease; RhD = Rhesus D; +ve = positive; −ve = negative; WBC = white blood cell; Hb = hemoglobin; plt = platelet; tx = transfusion; No. = number; PRBC = Packed red blood cell; U = unit.

**Table 2 diagnostics-13-00886-t002:** Distribution of RBC alloantibodies according to the specificities and clinical significance (*n* = 24).

RBC Alloantibody	Frequency, *n* (%)	Clinical Significance
**Number of alloantibodies**		
Single	20 (83.3)	
Multiple (≥2)	4 (16.7)	
**Antibody specificity**		
** Single alloantibody**		
Rhesus	9 (37.6)	
Anti-E	6 (25.0)	Yes
Anti-e	1 (4.2)	Yes
Anti-C	1 (4.2)	Yes
Anti-c	1 (4.2)	Yes
MNS	7 (29.2)	
Anti-M	1 (4.2)	Rarely
Anti-S	1 (4.2)	Yes
Anti-Mia	5 (20.8)	Rarely
Lewis	3 (12.5)	
Anti-Lea	2 (8.3)	Rarely
Anti-Leb	1 (4.2)	Rarely
No specificity	1 (4.2)	No
**Multiple alloantibodies**		
Anti-E and Anti-c	3 (12.5)	Yes
Anti-E and Anti-Fyb	1 (4.2)	Yes

No-specificity = presence of positive reaction in several of the screening and identification panel cells, but the pattern of reaction did not match any of antibody toward the tested antigen.

**Table 3 diagnostics-13-00886-t003:** Details on clinical data of alloimmunized CLD patients (*n* = 24).

No.	Age (Years)	Gender	Causes of CLD	ABO Group	Antibody Specificity	Transfusion	No. of PRBC/Plt Transfused (Unit)
1	47	M	NAFLD	A	Anti-E	No	0
2	61	F	Hepatitis C	A	Anti-E	No	0
3	27	M	Hepatitis C	B	Anti-E	Yes	2 Plt
4	73	F	Cryptogenic	O	Anti-E	Yes	2 PRBC
5	66	F	NASH	A	Anti-E	Yes	9 PRBC, 21 Plt
6	70	F	NASH	B	Anti-E	Yes	18 PRBC
7	43	F	NAFLD	A	Anti-c	Yes	4 PRBC
8	66	M	Hepatitis B	O	Anti-C	Yes	3 PRBC
9	32	M	AIH	A	Anti-e	Yes	3 PRBC
10	56	F	Hepatitis B	A	Anti-E, Anti-c	Yes	11 Plt
11	47	F	NAFLD	AB	Anti-E, Anti-c	Yes	6 PRBC, 2 Plt
12	58	M	NAFLD	O	Anti-E, Anti-c	Yes	12 PRBC, 16 Plt
13	59	M	NASH	O	Anti-E, Anti-Fyb	Yes	2 PRBC
14	65	F	Hepatitis C	B	Anti-S	Yes	12 Plt
15	52	M	Hepatitis C	B	Anti-M	No	0
16	68	M	Hepatitis C	A	Anti-Mia	No	0
17	52	F	Hepatitis B	A	Anti-Mia	No	0
18	70	F	Hepatitis B	O	Anti-Mia	No	0
19	69	M	NAFLD	B	Anti-Mia	No	0
20	41	M	Hepatitis C	O	Anti-Mia	Yes	3 PRBC, 2 Plt
21	66	F	Hepatitis B	B	Anti-Lea	No	0
22	61	M	Hepatitis B	A	Anti-Lea	No	0
23	51	M	Hepatitis C	B	Anti-Leb	No	0
24	49	M	Hepatitis C	B	No specificity	Yes	4 PRBC, 4 Plt

F = female; M = male; DM = diabetes mellitus; HPT = hypertension; SLE = systemic lupus erythematosus, COPD = chronic obstructive pulmonary disease; CKD = chronic kidney disease, NAFLD = non-alcoholic fatty liver disease; NASH = non-alcoholic steatohepatitis; AIH = autoimmune hepatitis; PRBC = Packed red blood cell; Plt = platelet; No-specificity = presence of positive reaction in several of screening and identification panel cells but did not match of any antibody toward the tested antigen.

**Table 4 diagnostics-13-00886-t004:** Factors associated with RBC alloimmunization among CLD patients by multiple logistic regression (*n* = 441).

Independent Variables	Crude OR (95% CI)	*p* Value	Adjusted OR	*p* Value
Age	0.99 (0.96, 1.02)	0.478	-	-
Gender				
F/M	1/1.62 (0.71, 3.71)	0.253	-	-
Ethnicity				
Non-Malay/Malay	1/2.04 (0.27, 15.59)	0.491	-	-
Causes of CLD:				
Viral Hepatitis	1			
MLD	0.70 (0.29, 1.72)	0.436	-	-
Cryptogenic	1.35 (0.17, 10.67)	0.778	-	-
ALD	0.00 (0.00, 0.00)	>0.950	-	-
AIH	0.43 (0.05, 3.69)	0.442	-	-
TLD	0.00 (0.00, 0.00)	>0.950	-	-
ABO blood group:				
O	1			
A	0.58 (0.20, 1.67)	0.310	-	-
B	0.73 (0.25, 2.17)	0.575	-	-
AB	1.76 (0.21, 15.06)	0.606	-	-
RhD Blood group:				
+ve/−ve	1/0.00 (0.00, 0.00)	>0.950	-	-
WBC (10^9^/L)	0.92 (0.83, 1.02)	0.122	0.56 (0.21, 1.58)	0.284
Hb (g/dL)	0.91 (0.75, 1.11)	0.353	-	-
Platelet (10^9^/L)	1.00 (0.99, 1.00)	0.134	1.08 (0.99, 1.15)	0.130
History of tx				
No/Yes	1/1.16 (0.50, 2.67)	0.726	-	-
No. of PRBC tx (U) *	1.03 (0.93, 1.14)	0.572	-	-
No. of Plt tx (U) *	1.06 (0.99, 1.13)	0.100	1.07 (0.67, 1.18)	0.103
Smoking (Yes/No)	1/1.66 (0.67, 4.08)	0.273	-	-
Alcohol consumption (No/Yes)	1/0.00 (0.00, 0.00)	>0.950	-	-

* = among transfused patients. F = female; M = male; CI = confidence interval; CLD = chronic liver disease; MLD = metabolic liver disease; ALD = alcoholic liver disease; AIH = autoimmune hepatitis; TLD = toxic liver disease; RhD = Rhesus D; +ve = positive; −ve = negative; WBC = white blood cell; Hb = hemoglobin; tx = transfusion; No. = number; PRBC = Packed red blood cell; Plt = platelet; U = unit.

## Data Availability

Data are contained within the article.

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
