# Peer review of "Red Blood Cell Alloimmunization and Its Associated Factors among Chronic Liver Disease Patients in a Teaching Hospital in Northeastern Malaysia"

_diagnostics, 2023, doi:10.3390/diagnostics13050886_

Round 1
Reviewer 1 Report
To authors
The study of transfusion allo-immunisation is an intrinsically interesting subject. In the present case, it is documented in a particular series of patients with chronic liver diseases of various origins. The series is too small for the observed results to reach statistical significance.
This study is well conducted and the article is very well structured and well documented bibliographically. The discussion is clear, pleasant to read and yet non-significant results are never easy to "discuss". The assessments (percentage greater than, smaller than, ...) - even if not statistically significant - are very skillfully compared with the conclusions of other studies on the same subject.
A few points deserve special attention; they are outlined below:
Page 2 line 60: "In Malaysia, multiple ethnicities have caused genetic heterogeneity....". In terms of immunisation rates, the difference between Malaysians and non-Malaysians is not significant. But it is known that genetic heterogeneity can sometimes be a problem because of the existence of specific phenotypes that are frequent in one population and rare in the other.
Page 2 line 82: among the socio-economic parameters characterizing the patients, smoking is taken into account in its own right, but - apparently - alcohol consumption is not, but it is included in the "clinical data". cfr. Table 1 ALD. Why is this?
Page 2 lines 91-94: From a methodological point of view, the algorithm used to screen for antibodies and then identify their specificity raises questions. Indeed, the method used to screen for anti-erythrocyte antibodies (3 cells in Saline + anti-globulin at 37°C) is less sensitive than the method used to confirm the presence of antibodies and identify their specificity, which uses the low ionic strength solution LISS (that is known to enhance agglutination) and red cells treated with papain.
Page 2 line 91-92 : « Antibody screening was performed using a three-cell panel at 37°C by saline indirect antiglobulin test, using a commercial screening cell reagent panel (ID-Diacell I-II-92 III, Asia, Bio-Rad) ».
Page 2 line 93-94 : « Sample with positive antibody screening proceeded with antibody identification using an eleven-cell panel, using low ionic strength solution (LISS) and en-zyme-treated cells (papain) at 37oC and AHG phase The entire test was performed using a commercialized cell panel by microcolumn gel agglutination method (ID-DiaPanel, Bio-rad) ».
Generally, the reverse is recommended: use a more sensitive method for screening than the confirmatory method, which should be more specific. This will help to identify possible false positive reactions of the screening test..
In this case, while the screening is suitable for the detection of clinically significant antibodies, there is a risk that it may not detect low titre or low affinity antibodies (e.g. in the case of recent immunisation).
This technological aspect should be discussed with the person in charge of the immuno-hematology laboratory performing the pre-transfusion tests.
Page 3 line 97: just specify that the phenotyping of patients was done using commercial reagents, and add : "following the manufacturer's recommendations".
In Table 2: what does "no specificity" mean? All reactions negative with panel cells? All reactions positive? All positive with papain-treated cells, but negative with anti-globulin? Explain.
Figure 2 : anti-mia should be written « anti-Mia » (and everywhere in the text)
Page 6 line 168: For the number of units of blood components transfused, would it not be more appropriate to look at the median rather than the mean?
Table 4: In the table header, "Crude OR" should have "Crude OR (IC95%)" in addition to "Crude OR (IC95%)" to indicate that it is the 95% confidence interval.
Discussion: the discussion is very full of relevant references on the same topic. It was not easy to discuss non-significant results, but the authors did it very cleverly.
Page 7 line 191 : « Differences in the allo-immunization rate among the studied population could be attributed to numerous factors such as patient demography, co-morbidities, number of transfusions, local transfusion practice, the sensitivity of the test method and heterogeneity of the population studied ». Just a small remark about the screening method which could be more sensitive as explained above, what can change the results.
Page 7 line 196: instead of "diagnoses", suggestion: replace with: "certain/some pathologies of the haematopoiesis or affecting the immune system such as auto-immunity, ...".
Page 8 line 206: As also observed in the study in reference 31, no anti-Kell was observed in the series. Even though the series is small and the K antigen is very rare in the study population, the high antigenicity of Kell (in Caucasians) deserves (perhaps) a small remark in this respect.
Page 8 line 210 : with 77% of patients at risk of anti-E immunisation, we understand the relative importance of this antibody among those detected.
Page 8 line 207 : Regarding "natural" or "immune" anti-E, it would have been interesting to specify whether the reactions observed with the cells of the panel were more intense in enzymatic medium than in the presence of anti-globulin. Another way to proceed would have been to treat the serum with DTT to eliminate the IgM component.
Page 8 line 215 : To reduce the rate of allo-immunisation in these at-risk patients, the authors « explain the need of extended Rh phenotype‑matched RBCs in patients requiring repeated transfusions to prevent Rhesus allo-immunization « .
In fact, it is not known whether in the series studied, patients actually received iso-pheno blood or not for CcEe Rh antigens. This is important and should be more clearly stated in the text. And even if anti-Mia antibodies are not known to be the most dangerous, Mia matching should perhaps also be taken into consideration. The consequences for such recommendations are - among other things - financial and workload, since they imply phenotyping not only of the recipient, but also of the donors to find Mi(a) neg donors. In the text, it is also not stated how and to what extend the donors are phenotyped.
Page 9 line 275 : « Our finding was inconsistent with the result from previous studies…. ». I find the wording too harsh as a non-significant result cannot - by definition - be compared to another one as it cannot be extrapolated either way. The most likely explanation is given in line 278.
Page 9 line 296: Why not include Mia matching in the matching recommendations for Asian patients?
Author Response
please see attaachment

Reviewer 2 Report
This study aims to identify the frequency and specificity of RBC alloimmunization among CLD patients in the Malaysian population.
This study showed a low prevalence of RBC alloimmunization among CLD patients treated the Malaysian population. However, majority of them developed clinically significant RBC alloantibodies, mostly from the Rh blood group. Therefore, they concluded that CLD patients requiring blood transfusion in the Malaysian population must at least be phenotyped for the Rh system and supplied with Rh phenotype specific blood for transfusion to prevent the Rhesus alloimmunization
Basically, the manuscript is well organized and the results are clear, with appropriate analysis and discussion. This analysis and the findings of this study thus provides useful insights for both clinicians and the scientific community. However, the author should consider addressing the following minor issues in this manuscript:
1) A description of the frequency of irregular antibodies in the Malaysian population is needed. This statistic will help us understand RBC alloantibody specificity among alloimmunized CLD patients as follow: Among these, the most common alloantibody identified belonged to the Rh blood 141 group (57.1), which was mainly contributed by anti-E (35.7%) and anti-c (14.2%), followed 142 by anti-Mia (17.8%) from the MNS blood group.
2) Detailed description of the differences in clinical course of CLD patients with RBC alloimmunization is needed, compared to CLD patients without RBC alloimmunization.
3) The unnecessary red letters need to be corrected.
